



# 1 A comprehensive integrated macroseismic dataset from
# 2 multiple earthquake studies

Andrea Tertulliani[1], Andrea Antonucci[2], Filippo Bernardini[3], Viviana Castelli[3],
Emanuela Ercolani[3], Laura Graziani[1], Alessandra Maramai[1], Martina Orlando[1,4],
Antonio Rossi[1], Tiziana Tuvè[5]
[1] Istituto Nazionale di Geofisica e Vulcanologia, Roma, Italy
[2] Istituto Nazionale di Geofisica e Vulcanologia, Milano, Italy
[3] Istituto Nazionale di Geofisica e Vulcanologia, Bologna, Italy
[4] Dipartimento di Scienze, Università Roma TRE, Rome, Italy
[5] Istituto Nazionale di Geofisica e Vulcanologia, Osservatorio Etneo, Catania, Italy
*Correspondence to: andrea.tertulliani@ingv.it*
**Abstract**
Each Italian earthquake included in the Italian Parametric Catalogue (CPTI) is based on a single study,
with its database stored in the Italian Macroseismic Database (DBMI). DBMI collects macroseismic
intensity data from approximately 5,000 Italian earthquakes. However, for the same events, numerous
studies have been independently carried out over the years in the literature whose data have not been
incorporated into the DBMI. By consolidating all available data for each event, it is possible to
significantly enhance the dataset used for hazard assessments and the reconstruction of local seismic
histories. This approach would make studies of individual events much more robust and comprehensive.
The objective of this work is to propose the integration of different macroseismic datasets for individual
events by identifying criteria that can effectively merge a large number of intensity data points.
A total of 45 Italian earthquakes with data from multiple sources were identified and reassessed through
a rapid review process. This effort has resulted in the creation of a new dataset, substantially increasing
the number of Macroseismic Data Points (MDP) for the earthquakes covered by this study compared to
those in DBMI15 (from 2,892 to 9,328 MDPs). Consequently, the macroseismic distributions for these
45 events have become more detailed, robust, and extensive.
## 31 1 Introduction
In the last few decades, a huge amount of information on the seismic history of Italy was produced,
contributing to the compilation of the current seismic catalogue, the Parametric Catalogue of Italian
Earthquakes - CPTI15 (Rovida et al., 2020; 2022a). CPTI15 lists 4894 events located in the entire Italian
territory and neighboring areas from 1000 AD to 2020, and is fed by the Italian Macroseismic Database
- DBMI15 (Locati et al., 2022), which contains over 120,000 Macroseismic Data Points (MDPs) related
to more than 3200 earthquakes. The single MdP is the geographical site where the effect of the ground
shaking of an earthquake has been observed, synthetically described with a macroseismic intensity value.
Indeed, each of those data points is provided by geographical coordinates and an intensity value. This
huge amount of data comes from approximately 190 studies produced over time by the scientific
community and dedicated to one or more earthquakes. In many cases, several studies are available in the
literature on the same earthquake. Such studies, produced at different times by different authors and with
distinct research methods, ensure a multiplicity of views and types of information that is, in itself, a great
contribution to the progress of scientific knowledge and a valuable help for potential future research.
To keep abreast of this impressive scientific production, in 2017 the Italian Archive of Historical
Earthquake Data (ASMI) was created (Rovida et al., 2017; Rovida et al., 2024). Since 2017, ASMI has



been continuously implemented, collecting many references of interest, related not only to the thousands
of earthquakes included in the CPTI15 catalogue, but also to earthquakes that are below the energy
thresholds set for inclusion in CPTI15 (intensity 5 and/or magnitude 4). To date, ASMI stores about 460
different data sources related to a total of about 6700 earthquakes.
The epicentral parameters of each event listed in the CPTI-DBMI catalogue are based on a single
reference study (hereafter "preferred"), selected from among those collected in ASMI, with criteria based
on the intrinsic quality of the study itself.
A screening of all the studies available for different earthquakes has pointed out that "preferred" studies
are not always those that provide the largest number of MDPs, nor the most recent or up-to-date ones.
Indeed, in several cases, studies of the same earthquake by different authors can produce different
datasets, in terms of the number of collected MDPs, the geographic distribution of the same, the adopted
macroseismic scale, or methods used for collecting data.
It is important to note that the Italian Macroseismic Database does not include all the MDPs available
for a given earthquake, but only those reported in the study preferred by the catalogue for that earthquake.
This means that any MDPs available outside the preferred study, run a great risk of being overlooked
and ignored in further analysis of that same earthquake. This would be a great loss because, as was
recently highlighted by a detailed analysis (Orlando et al., 2024), these different datasets are, in many
cases, complementary to each other.
The integration of different datasets has been occasionally carried out in recent years (Graziani et al.,
2017; Tertulliani et al., 2018), but so far, no general criteria for systematic applications have been
established. The goal of this work is to verify if it is possible to integrate different datasets in one intensity
compilation quickly and efficiently while retaining the good quality of intensity assessments, without
conducting a thorough and time-consuming revision of each earthquake. This operation would allow us
to systematize a considerable amount of data under-used or completely disregarded in previous studies.
The unquestionable advantages of such an operation are: (i) enhancing the macroseismic database by
adding a large number of previously overlooked MDPs, thereby improving and expanding the seismic
histories (i.e., the list of effects observed in a place through time) of many locations; (ii) improving the
knowledge of single earthquakes, thus providing the catalogue with more robust and reliable datasets;
(iii) enriching the available datasets in intensity values from both MCS and EMS-98 scales.
To this end, we selected from CPTI15 a set of 45 Italian earthquakes for which multiple datasets coming
from different macroseismic sources are available in ASMI. We built a new dataset consisting of 9328
MDPs, expressed both in the MCS and EMS-98 scale (Tertulliani et al., 2024) that may be incorporated
into the CPTI-DBMI database. This paper describes the input data that were used and the methodology
adopted for building the new dataset. The exposition of some case studies and an analysis of the results
and contents are also included.
**2 The macroseismic intensity**
Macroseismic intensity is a measure of the effects of an earthquake, as perceived, experienced, and
recorded by people, buildings, and the natural environment at specific sites. While magnitude is a
quantification of the energy released by an earthquake at its source, macroseismic intensities summarize
how the shaking produced by that energy release was felt and the consequences it produced at different
points on the earth's surface. Macroseismic intensity is defined according to discrete scales, whose
degrees are related to standard descriptions or scenarios of seismic effects. The most common
macroseismic scales are the MCS (Mercalli-Cancani-Sieberg, Sieberg, 1932), the MMI (Modified
Mercalli Intensity, Wood and Neumann, 1931) and the MSK (Medvedev-Sponahuer-Karnik, Medvedev
et al., 1965). In the last few decades, the recent EMS-98 (European Macroseismic Scale, Grunthal, 1998)
has been gradually taking over on earlier scales, particularly in Europe.
The information needed to assess the macroseismic intensities of recent earthquakes can be gathered in
two main ways: either through questionnaires filled in by inhabitants in the affected areas (either directly
or via online forms); or through field surveys, carried out by experts, aimed at collecting evidence of
damage and environmental changes (e.g. landslides, ground fissures, etc.). The assessment of



macroseismic intensities has always been a field reserved for expert seismologists, but it is undeniable
that some subjectivity of interpretations is implicit in the process. Accordingly, in the past few decades,
algorithms have been created with the aim of reducing subjectivity, particularly in processing large
masses of data from crowdsourced macroseismology (Gasparini et al., 1992; Quitoriano and Wald, 2020;
Sbarra et al. 2010). In the case of historical earthquakes (i.e. those for which intensities must be assessed
secondhand, from descriptive evidence), intensity evaluation is carried out after a careful screening and
study of historical sources, by means of a process of translating original accounts and information into
diagnostic elements.
**3 Input data**
We selected from CPTI15 (Rovida et al. 2022) 45 earthquakes with Mw ranging from 2.5 to 5.8, dated
from 1985 to 2006 and located over the whole Italian territory (Figure 1). The selected earthquakes, 26
of which occurred in the Etna volcanic region, are supported by a total of 2896 MDPs (Table A1). For
these earthquakes, several different datasets are available on ASMI (Rovida et al. 2017; (Rovida et al.,
2017; Rovida et al., 2024), provided by various kinds of studies (reports of direct field surveys, data
collections through questionnaires, and preliminary or final reviews). In some cases, other kinds of
datasets are also available, such as data collected by sending questionnaires to schools or by individual
macroseismic studies (i.e. Guidoboni et al., 2018). Using such a variety of macroseismic studies to assess
intensities, means to deal with inhomogeneous data, collected by different research teams, at different
times, with different means and criteria, and using different macroseismic intensity scales.
To make a couple of examples, some studies provide intensity datasets georeferenced at a municipal
scale, i.e. they provide for each municipal territory a single intensity degree. This data can be based either
on one scenario of effects detected in a single inhabited site (e.g. the main locality of the municipality),
or on the cumulation of scenarios detected in as many inhabited sites (*hamlets*) constituting the municipal
territory. Other studies provide more detailed datasets, with intensity degrees assessed at the scale of
*hamlets*.
Regarding intensity scales, until the year 2000, the MCS scale was mostly used in Italy. Subsequently, it
was gradually supplanted by the adoption of the EMS98 scale, particularly for direct field surveys.
Below, a brief description of the most recurring input datasets used for the present work is shown.



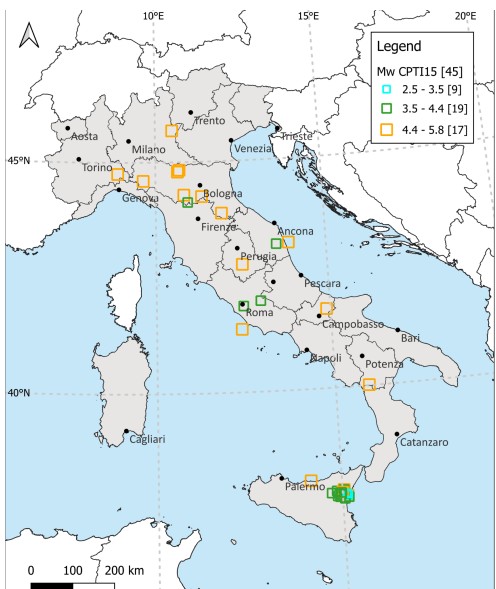

**Figure 1: Distribution of the selected earthquakes.**

### 3.1 The ING/INGV Macroseismic Bulletin

The Macroseismic Bulletin of Istituto Nazionale di Geofisica e Vulcanologia - INGV (ING before 2000) is the main source of macroseismic data for most of the medium-to-low energy earthquakes that occurred in Italy from 1980 to 2009.

In 1978, the Istituto Nazionale di Geofisica (ING) signed an official agreement with the General Command of the Italian Carabinieri Corps to establish a dense network of correspondents capable of providing a continuous service for the collection of macroseismic observations in the aftermath of earthquakes (Favali et al., 1980). When an earthquake occurred, questionnaires were sent by ING to the Carabinieri stations located in a large area around the epicenter. Filled questionnaires were returned to ING (Figure 2), where a team of experts processed them and derived estimates of the macroseismic intensities (e.g. Spadea et al., 1983; 1984; 1985). In the following years, the network expanded to include other public bodies, such as the Italian Municipalities and Forest Guard stations, in order to increase the quantity and quality of the collected information. In the early 1990s, the network of correspondents consisted of more than 13,000 observation points, covering the entire country (Gasparini et al., 1992). This data collection service remained in operation until 2009.


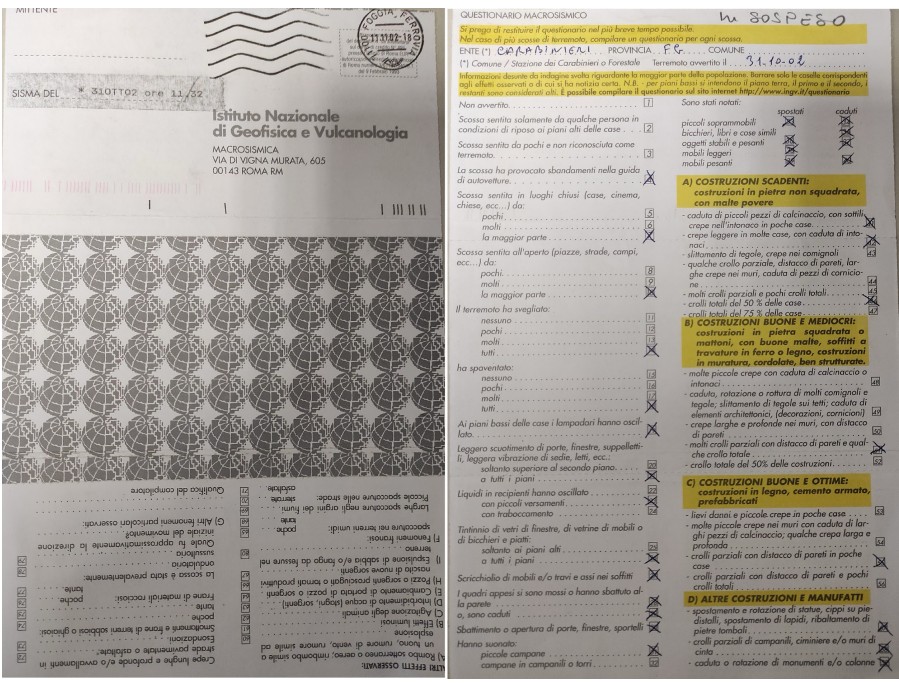

**Figure 2: Example of a hard copy questionnaire of the ING Macroseismic Bulletin used during the 1990s.**

The threshold earthquake magnitude for data collection was set at approximately magnitude 3.0, to gather
information on medium-to-low energy earthquakes for which no field surveys would be carried out. The
questionnaires included numerous questions on how the earthquake was perceived by people, its effects
on objects inside buildings, damage to buildings of different types and also environmental effects.
Until 1988, the questionnaires were based on the MSK and the MCS scales, and intensity was assigned
according to both. From 1988 onwards they were based on the MCS scale only. The information gathered
from questionnaires for each earthquake was used by the ING staff to assess macroseismic intensity for
each site, employing an algorithm based on weighted means, in order to minimize subjectivity in the
estimation of intensities (Gasparini et al., 1992). The resulting macroseismic data were published yearly
in a Macroseismic Bulletin as a list of MDPs for each earthquake (e.g., Gasparini et al., 1994; 2003;
2011). The Macroseismic Bulletins used as a source in this study, is one of the main data sources
employed by the scientific community to study Italian seismicity and for compiling the DBMI15-
CPTI15. Over the entire operational period of the Bulletin, intensity data for over 2400 earthquakes were
collected, 392 of which have been considered as main ref (preferred reference) in the DBMI15-CPTI15,
contributing with more than 35,000 MDPs. It should be stressed that, unlike direct surveys, a vast
majority of data contained in the Bulletin are characterized by low-intensity values.

**3.2 Direct field-surveys**
Some of the earthquakes considered in this paper are characterized by studies (and related datasets)
resulting from macroseismic surveys carried out in the field by teams of experts. Usually, direct
macroseismic investigations in earthquake-affected areas are performed for earthquakes exceeding a
given magnitude threshold (Bottari et al., 1980, Camassi et al., 2008, 2009). They produce data that,
having been collected by specially trained personnel, have a higher level of reliability than those collected





through questionnaires. This latter circumstance was taken into account when establishing the criteria
adopted in this study for merging the different datasets.
The goal of macroseismic field surveys is to assess intensity at a specific locality by direct observation
of the effects produced by an earthquake in that locality. These effects can be either transient (those on
people and objects) or permanent (building damage). When the scenario shows very minor and sporadic
damage, data collection focuses more on transient effects, gathered both through press reviews and,
above all, by interviewing the affected populations: people describe how they perceived the shaking and
where (i.e. whether indoors or outdoors), and the effects they observed on household objects
(oscillations, falls, breakages). Conversely, when widespread damage ranging from moderate to severe
occurred, the survey is mainly focused on building damage and may include vulnerability assessments
of the whole building stock. The field-collected data serve as raw inputs, which, when analyzed according
to the guidelines of the adopted macroseismic scale, allow the intensity to be estimated (Grunthal, 1998,
Molin 2009).
Over the years, direct survey techniques have evolved, both because influenced by the adoption of
different macroseismic scales and also to enhance objectivity in the investigation (Del Mese et al. 2023).
As a result, macroseismic data derived from direct field surveys carried out at different times and with
different methods, can show inconsistencies and inhomogeneity. Such inhomogeneity can be mainly
ascribed to the adoption of different macroseismic scales or even to the different geographical extent to
which the survey was performed (municipality level vs hamlets level).
Generally speaking, however, regardless of the period in which they were conducted, the results of direct
field investigations are to be considered among the most reliable macroseismic data ever.
Due to time constraints and issues related to the availability of skilled personnel to deploy in the field,
data from surveys, while detailed in the epicentral areas, often have a rather limited extent in the far-
field, in contrast with data collected with indirect techniques. This is why data derived from direct field
surveys are often incomplete in the far-field. Therefore, for a given earthquake, these studies are more
suitable to be integrated with other studies that provide more complete far-field datasets.

**3.3 Other kind of datasets**
Our study includes 26 earthquakes located in the Etna Volcano region (Sicily), whose data come from
the Macroseismic Catalogue of Etna Earthquakes (CMTE, Azzaro and D'Amico, 2014). This catalogue
- the most updated collection of earthquakes existing related to this volcanic area - lists 1,874
earthquakes, occurring between 1633 and 2023, including both fore- and after-shocks, 220 of which
exceed the damage threshold. To date, the related macroseismic database contains 9274 MDPs with an
associated intensity dataset assessed according to the EMS-98 scale. The compilation of CMTE is the
result of the analysis of about 200 primary sources (scientific papers, bulletins, newspapers, archive
documents, and direct surveys), providing a complete and homogeneous dataset to investigate local
seismicity over the last 4 centuries.
In the 1980s and 1990s, some Italian seismological agencies collected macroseismic information by
means of questionnaires distributed to schools, to gather dense feedback from students (Esposito et al.,
1988; Tertulliani and Donati, 2000). These data are plentiful but often of poor quality, due to the
impossibility of checking the competence of the compilers. Anyway, at least for some earthquakes, these
are the only data available for intensity assessment.
**4 Methodology**
Unifying the results of different macroseismic studies cannot be achieved by a mere combination of
intensity values. First, it is necessary to identify homogenisation criteria to optimise the quantity and
quality of data. As already mentioned, the differences depend on the different methods of data collection
(which vary according to historical periods), the macroseismic scale used, and the way it was used. The

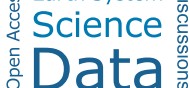

studies associated with the earthquakes analyzed in this work provide datasets that differ both in the
number of MDPs and in the intensity values assessed to each point. Sometimes, different studies list the
same localities, either assessing the same intensity value or not. Conversely, only one of the available
studies can report some or many localities for a given earthquake. In fact, by comparing the datasets of
each earthquake we can find the following data layouts:

- localities that are included in all the available studies, with identical or different assigned intensities, in MCS scale;
- localities that are included in only one of the available studies, with MCS or EMS-98 intensity;
- localities that are included in all the available studies, with identical or different assigned intensities, in both MCS and EMS-98 scales.

To accomplish our task efficiently and systematically, it was necessary to establish transparent criteria
and to make a few assumptions about the nature of the data to be processed.
Taking a cue from recent experiences in macroseismology (Musson et al., 2010; Del Mese et al., 2023;
Castellano et al. 2018; Bernardini and Ercolani, 2023), we adopted some guidelines that, we believe, can
be applied to the entire datasets being compared.

Firstly, we defined the following initial criteria:

a. Localities with intensity value (I) in the EMS-98 and MCS scales assigned after a field survey have been included in the new dataset without further check, assuming that values assessed by expert personnel are robust and reliable.
b. Localities for which a single study assesses $I \geq 5$, not resulting from a field survey, have been reviewed, whatever scale is used.
c. Localities for which different studies assess two intensity values $\geq 5$ on the same scale but with a difference greater than or equal to 1 degree have been reviewed; such an important difference in intensity requires further evaluation to assess which diagnostics led to different estimates.
d. Localities in which different studies assess two intensity values $< 5$ on the same scale but differing each other by a half degree of intensity (i.e., $I_1 = 4\text{-}5$ and $I_2 = 4$), the integer value between the two (i.e., $I = 4$) has been assigned, according to the EMS-98 guidelines.
e. Localities for which a single study reports $I < 5$ have been included in the new dataset without further verification. For lower intensity levels, where the estimation relies on transient effects, the literature (e.g., Musson et al., 2010; Sbarra et al., 2020) indicates that MCS and EMS-98 estimates are roughly equivalent. Therefore, regardless of the scale, the intensity value can be considered reliable for both the MCS and EMS-98 datasets.

In the case of localities with intensity from different scales:

1. $I = 5\text{-}6$ MCS has been considered equivalent to 5 EMS; this assumption is based on the different definitions of intensity degrees 5 and 6 in the two scales: the onset of damage to buildings is expected at intensity degree 5 in the EMS-98, and at intensity degree 6 in the MCS scale.
2. $I < 5$ MCS has been considered equal to the same EMS-98 value; on this assumption see criterion *"e"* above.

In addition, in all cases where the intensities assigned to localities in different studies have shown
significant differences or when the available data are doubtful or lacking, a revision has been done.

It should be noted that, very often, the raw data collected either through direct surveys or through
questionnaires, is aimed at defining an intensity estimate according to the MCS scale. However, in order
to assign EMS-98 intensity from these data, we had to make some reasonable assumptions to compensate



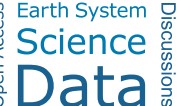

for the lack of information on building vulnerability classes, damage grades and observed damage
frequency. To overcome this criticality, the information contained in the questionnaires can be helpful.
These latter, though not required to fulfill EMS-98 diagnostics, were meant to assess intensity in MSK,
from which EMS-98 directly derives. By a careful examination of the answers to questionnaires, we were
able to obtain a rough estimate of vulnerability classes and damage grades.
**5 Case studies**
Three significant examples of this revision process are represented by the earthquakes that occurred in
1987 in the Marche region (Central Italy), in 2002 in the Molise region (Southern Italy), and in the Etna
volcanic area.
The earthquake of July 3, 1987 (https://emidius.mi.ingv.it/ASMI/event/19870703_1021_000), with a
moment Magnitude (Mw) of 5.06 and a maximum epicentral intensity (Imax) 7 MCS, underwent a
significant revision based on two main sources: the ING Macroseismic Bulletin (Gasparini et al., 1988),
which is the preferred study of DBMI15-CPTI15 and contains 359 MDPs (Figure 3a), and the study by
Monachesi and Raccichini (1987) that provides 36 MDPs coming from direct field-survey. The analysis,
which involved 78 specific checks, led to substantial modifications of the original datasets (Figure 2b).
In particular, 7 MDPs reported in the ING Macroseismic Bulletin were excluded from the Tertulliani et
al. (2024) dataset as the effects initially attributed to this event were subsequently linked to the
earthquake of July 5 of the same year, which occurred close to the felt area of the studied event
(https://emidius.mi.ingv.it/ASMI/event/19870705_1312_000). Due to the absence of original
questionnaires and the presence of contradictory information, it was not possible to assign an intensity
value for six localities. The revision work also identified questionnaires related to six previously
unconsidered localities and added them to the intensity data now consisting of 373 MDPs (Figure 3b).
Furthermore, the maximum intensity, initially estimated as 7 MCS scale in the ING Macroseismic
Bulletin, in this study has been revised to 6-7 MCS and 6 on the EMS-98.
We also calculated the macroseismic magnitude MwM with the algorithm proposed by Gasperini et al.
(1999; 2010) using the resulting intensity data (Tertulliani et al., 2024). The estimated MwM results
equal to 4.94 for the event that occurred on 3 July 1987 and differs by 0.34 units from those of the Italian
catalogue (i.e., MwM 5.28). This difference can be attributed to the downward reassessment of the
intensities of several localities.
The second significant case study concerns the October 31, 2002, Molise earthquake
(https://emidius.mi.ingv.it/ASMI/event/20021031_1032_000), with Mw of 5.74 and Imax of 8-9 MCS.
For this event, the data from the preferred study of DBMI15-CPTI15 (Bosi et al., 2002), with 51 MDPs,
and the INGV Macroseismic Bulletin (Gasparini et al., 2011) with 790 MDPs, were analyzed (Figure
3c). Bosi et al. is a technical report compiled after the direct survey in the aftermath of the earthquake,
focusing on near-field effects, while Gasparini et al. extend the data collection in the far field. Our
revision, which required 168 specific checks, highlighted many necessary changes in the intensity
evaluation: for a specific locality, the intensity was reduced after a careful analysis of both photographic
documentation and field survey descriptions. The final result of this revision led to an increase in the
number of MDPs to 798 (Figure 3d), compared to the reference study in DBMI15-CPTI15 (51 MDPs),
integrating the epicentral data with the observed effects in the far field. In this case, the macroseismic
magnitude (MwM 5.27) is very similar to the macroseismic magnitude reported in CPTI15 (i.e., MwM
5.33), however the error associated to the new estimate is significantly reduced, from +- 0.23 to 0.04.



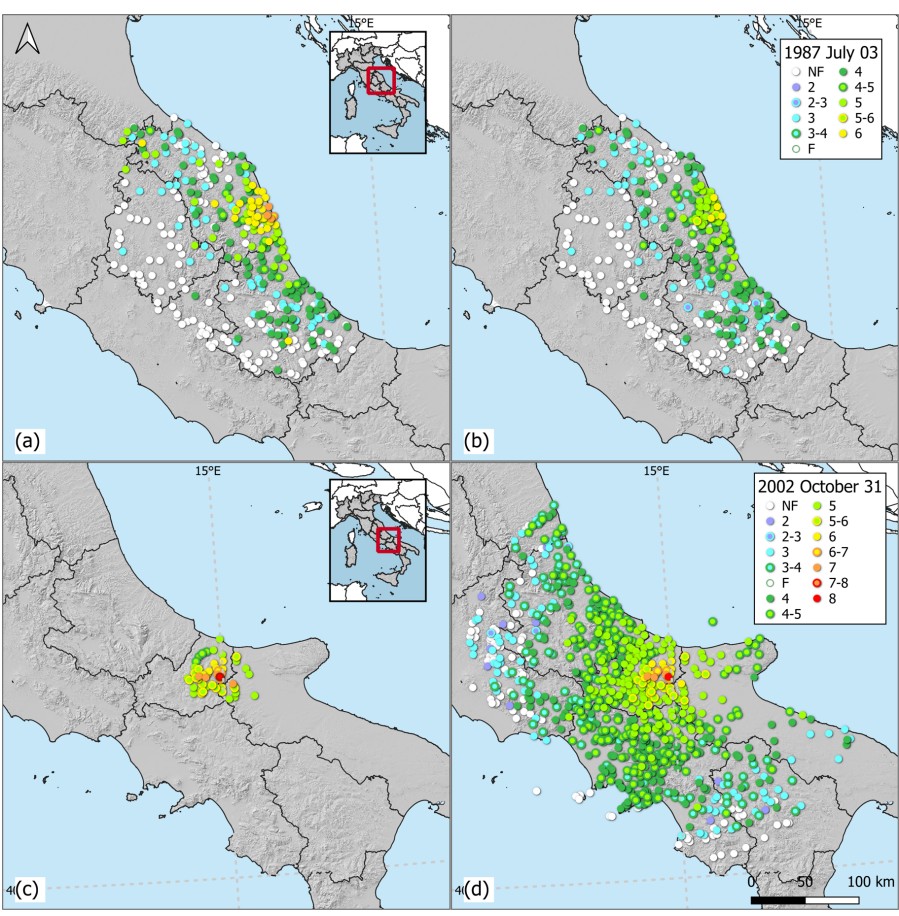

**Figure 3: Intensity distribution of 1987 July 3 (a) and 2002 October 31 earthquakes (c) as reported in DBMI15 in the MCS scale and this study (b) and (d) in the EMS-98 scale, respectively.**

The third example is the earthquake recorded in the Etna area, near Piano Provenzana, on October 27, 2002 (https://emidius.mi.ingv.it/ASMI/event/20021027_0250_000), with Mw of 4.84 and maximum intensity equal to 8 EMS-98 (Figure 4a). The revision of this event has been based on two sources: the direct field-survey by Azzaro et al., 2006, which is the preferred study in DBMI15-CPTI15, providing 17 MDPs, and the INGV Macroseismic Bulletin (Gasparini et al., 2011), which lists 101 MDPs.

The analysis included 54 specific checks and for 7 of these, only the intensity data from the direct survey was available. Additionally, 4 MDPs reported in the INGV Macroseismic Bulletin were excluded by Tertulliani et al. (2024), because the revised questionnaires were unreliable.

In this case, as well, the revision work led to an important increase in the number of MDPs, which now totals 106 MDPs (Figure 4b), and the maximum intensity has been confirmed as 8 EMS-98.



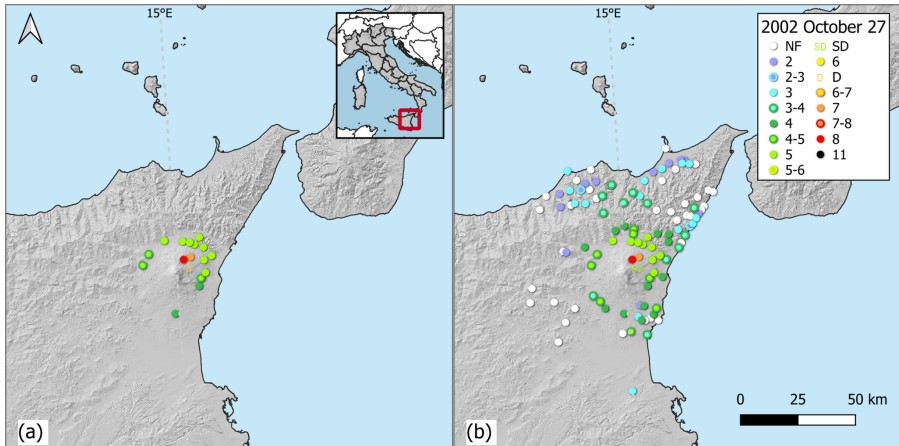

**Figure 4: Intensity distribution in the EMS-98 scale of the 2002 October 27 earthquakes as in DBMI15 (MDP set by Azzaro et al., 2006) (a) and in this study (b).**

## 6 Results

This work allowed us to reconstruct a new complete dataset (Tertulliani et al., 2024) for 45 Italian earthquakes that occurred from 1985 to 2006. It represents the final result of a systematic harmonization and homogenization of both intensity data and geographical coordinates for each locality. This task was performed by a careful check of about 2000 macroseismic questionnaires (see Section 3.1) and of many other sources of various kinds. During this work, we were also able to correct several misinterpretations in the previous assessment of intensity verifying the accuracy of the match between the effects produced and the assigned intensity. In this respect, 53 MDPs contained in the macroseismic bulletins were discarded from Tertulliani et al. (2024): 46 MDPs had incorrectly filled out questionnaires providing ambiguous information, while 7 MDPs were mistakenly referred to one event instead of another.

For the 45 earthquakes studied (Tertulliani et al. 2024), the number of intensity data has increased from 2892 MDP, currently included in DBMI15, to 9328 MDP as the final result of the present work. As Figure 5 and Table A1 show, for ten of the considered earthquakes the number of MDPs increased more than 500% with respect to those presently collected in DBMI15, while for the other 25 earthquakes, the increase in the number of points was greater than 100%.


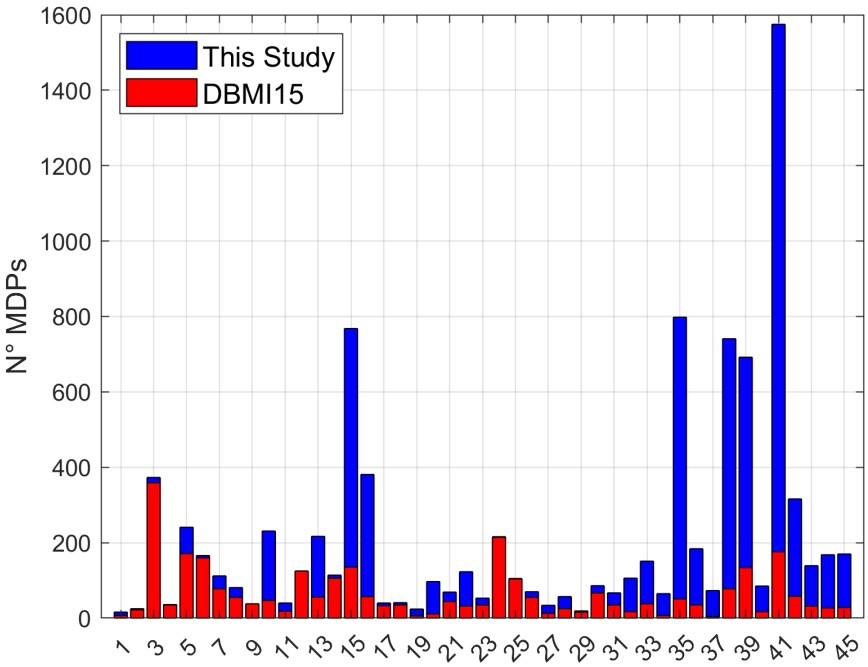

**363**

**364**    **Figure 5: Number of MDPs of the selected earthquakes as reported in DBMI15 (red bars) and in this study**
**365**    **(blue bars). In the horizontal axis the progressive number of the studied earthquakes as reported in Appendix**
**366**    **A.**

**367**

**368**    Furthermore, the intensity data contained in Tertulliani et al. (2024) are now provided in both MCS and
**369**    EMS-98 scales. Figure 6 shows the data distribution as a function of each intensity degree showing that
**370**    the frequency of each intensity class is higher than those reported in DBMI15 for both macroseismic
**371**    scales. In particular, Figure 6a shows that, after the revision, the number of data is 105 and 845 for
**372**    intensities 6 EMS-98 and 5 EMS-98 respectively, increasing respectively of 320 % and 754 %, whereas
**373**    Figure 6b shows that the number of data in the MCS scale is equal to 993 for intensity 3 and 1370 for
**374**    intensity 4-5, that correspond to an increase of 397% and 512% respectively. In addition, for intensity 5-
**375**    6 the number of total data is slightly different between the two scales: 246 MDPs are present for MCS,
**376**    and 120 for EMS-98. This discrepancy is due to the different diagnostics used by the two scales for the
**377**    degrees 5 and 6.
**378**    This huge increment of MDPs with intensity < 6 means, unlike previously, that the macroseismic data
**379**    for many of the studied earthquakes are now representative of the entire impact area of the event, from
**380**    the epicentral area to the far field, where the earthquake was just slightly felt. In fact, the increase in the
**381**    number of low-intensity data is complemented by the significant amount of data related to localities
**382**    situated at great epicentral distances. Figure 7 shows that, for the studied events, for I < 5 the number of
**383**    data placed at distances > 100 km is significantly higher than that contained in DBMI15. Indeed,
**384**    considering intensities ≥ 2, Tertulliani et al. (2024) provide 1157 MDPs located at epicentral distances
**385**    > 100 and 78 MDPs at distances > 200 km, with respect to 171 MDPs and 9 MDPs included in DBMI15
**386**    for the same distances.
**387**    As a result of the revision, the total amount of data contained in the dataset is referred to 5027 Italian
**388**    localities. Out of these, 129 were not reported in DBMI15, while 3151 localities, related to the examined
**389**    earthquakes, have been assigned a new intensity value.
**390**    Going into detail, the earthquake that showed the greatest increase in the amount of data is the one that
**391**    occurred       in       Northern       Italy       on       24       November       2004       (ID       41:
**392**    https://emidius.mi.ingv.it/ASMI/event/20041124_2259_000, last access 28 October 2024), for which,



Earth System
Science
Data

thanks to the results of our study, a total of 1575 MDPs are now available, compared to 176 MDPs
currently included in DBMI15.

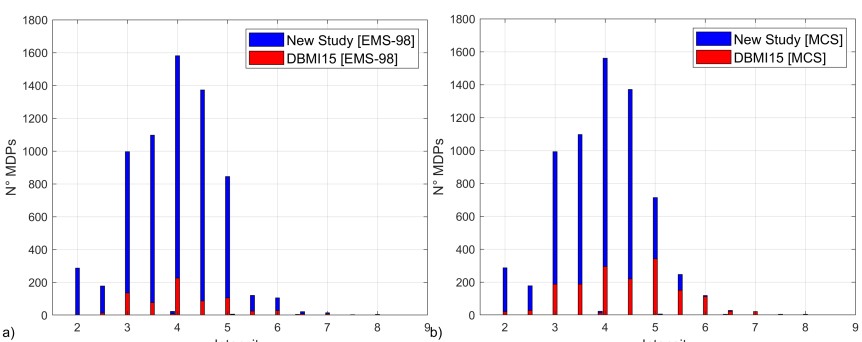

a)                                             b)

**Figure 6: Number of MDPs as a function of each intensity degree in EMS-98 (a) and MCS (b) provided in this**
**study (blue bars) and in DBMI15 (red bars).**

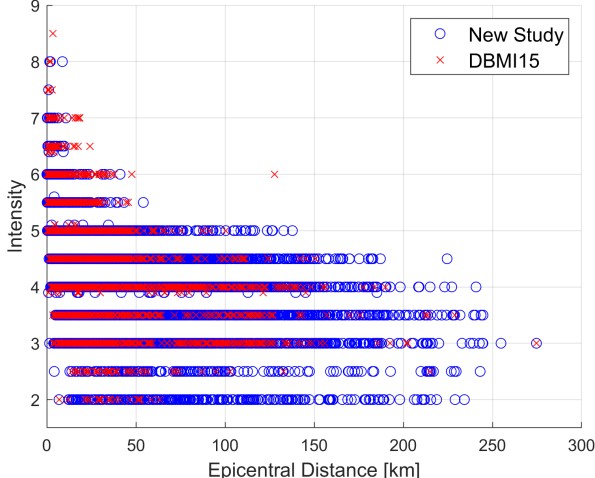

**Figure 7: Epicentral distance vs intensity class of the data contained in the new study (blue) and in DBMI15**
**(red).**
**7 Data availability**
The integrated dataset (Tertulliani et al., 2024) is available at
https://doi.org/10.13127/macroseismic/teral024 and it is released under a Creative Commons Attribution
4.0 International (CC BY 4.0) license. The data file is downloadable in both Portable Document Format
(.pdf ) and MS Excel (.xlsx) formats through the ASMI web portal (https://emidius.mi.ingv.it/ASMI/,
last access: 10 December 2024). The downloadable spreadsheets contain the list of 9328 MDPs, as
described in the previous sections, together with the associated references and format description of the
contained field. The dataset is also available through ASMI's web services



(https://emidius.mi.ingv.it/ASMI/services/), according to the standards of the International Federation of
Digital Seismograph Networks (fdsnws-event) and the Open Geospatial Consortium, in particular the
Web Feature Service (OGC WFS) and the Web Map Service (OGC WMS).
**8 Conclusions**
In this work we made a revision aimed at making a new and complete dataset for several earthquakes
with the goal of including all MDPs coming from different macroseismic studies. In this respect, we
identified several criteria aiming at integrating different datasets into an unique reliable intensity
compilation in a fast and robust way. Tertulliani et al. (2024) represents the result of this compilation of
a total of 9328 MDPs related to 45 Italian earthquakes that occurred from 1985 to 2006, expressed in the
EMS-98 and MCS macroseismic scales. This dataset allows to strongly increase the total number of data
available with respect to those already contained in DBMI15 (from 2892 to 9328 MDPs) and to make
the macroseismic distribution of the 45 events more solid, robust, and extensive.
In addition, the increment of the MDPs has allowed to broaden the spatial distribution of the intensity
observations, making it possible to include many data from the far field of the considered events. This,
arguably, has positive influences on the parameterizations of the events themselves, which are now based
on more exhaustive datasets.
An important finding of our study has been the improvement of the "seismic histories" (i.e., the list of
earthquakes experienced through time by a locality) of 3151 Italian localities. Indeed, for many of the
localities affected by the examined earthquakes, an intensity value was assigned for the first time as a
result of our study: up to now, these places were not known to have experienced seismic events. As a
relevant fact, it has to be underlined that the 45 analysed earthquakes occurred in an era in which
instrumental data already had high reliability. This offers the possibility of using this large amount of
new intensity data for many seismological purposes, such as calibrating the methods for deriving
earthquake parameters, the intensity prediction equations (IPE) s, and the ground-motion-to-intensity
conversion equations (GMICE).
The concept of conducting a review based on objective criteria makes this methodology broadly
applicable to other earthquakes, enabling a more efficient and systematic enhancement of knowledge
about Italian seismicity. This approach avoids the need for exhaustive earthquake re-evaluation and
focuses instead on addressing cases where datasets exhibit potential inconsistencies or nonhomogeneity.
In our analysis, only 1783 out of 9328 MDPs were re-examined, demonstrating the efficiency of the
review process and its ability to streamline efforts without compromising reliability. The proposed
methodology is particularly effective for the rapid yet reliable updating of medium-low earthquakes,
which are characterized by a vast amount of low-intensity data. Such kinds of earthquakes are not only
numerous but also critical for understanding regional seismic activity. While they often do not cause
major damage, they are significant because they can still generate notable shaking, leading to localized
damage and frightening among the population. Consequently, their study is essential for refining
historical seismic histories and contributing to enhancing the seismic hazard of a given area.



**Appendix A**
Table A1. ID, ASMI link, Data, and Epicentral Area of the 45 selected earthquakes with their number of
MDPs reported in DBMI15 (MDP DBMI15), number of data revised (MDP Rev), and total number
provided by this study (MDP This Study).

| ID | ASMI ID | Date | Epicentral Area | MDP DBMI15 | MDP Rev | MDP This Study |
|---|---|---|---|---|---|---|
| 1 | https://emidius.mi.ingv.it/ASMI/event/19850815_1858_000?page=2 | 1985 08 15 | Parma Apennines | 7 | 6 | 16 |
| 2 | https://emidius.mi.ingv.it/ASMI/event/19870202_1608_000?page=2 | 1987 02 02 | Eastern Sicily | 22 | 3 | 25 |
| 3 | https://emidius.mi.ingv.it/ASMI/event/19870703_1021_000?page=2 | 1987 07 03 | Marche Coast | 359 | 78 | 373 |
| 4 | https://emidius.mi.ingv.it/ASMI/event/19870813_0722_000?page=2 | 1987 08 13 | Etna_Maletto | 35 | 1 | 36 |
| 5 | https://emidius.mi.ingv.it/ASMI/event/19880108_1305_000?page=2 | 1988 01 08 | Pollino | 171 | 53 | 243 |
| 6 | https://emidius.mi.ingv.it/ASMI/event/19880315_1203_000?page=2 | 1988 03 15 | Reggiano | 160 | 46 | 166 |
| 7 | https://emidius.mi.ingv.it/ASMI/event/19890129_0730_000?page=2 | 1989 01 29 | Etna_Codavolpe | 78 | 34 | 112 |
| 8 | https://emidius.mi.ingv.it/ASMI/event/19890727_1508_000?page=2 | 1989 07 27 | Etna_Caselle | 55 | 16 | 81 |
| 9 | https://emidius.mi.ingv.it/ASMI/event/19911215_2000_000?page=2 | 1991 12 15 | Etna_Southern side | 38 | 18 | 38 |
| 10 | https://emidius.mi.ingv.it/ASMI/event/19930626_1747_000?page=2 | 1993 06 26 | Madonie Mountains | 47 | 28 | 231 |





| | | | | | | |
|---|---|---|---|---|---|---|
| 11 | https://emidius.mi.ingv.it/ASMI/event/19950210_0815_000?page=2 | 1995 02 10 | Etna_Western side | 18 | 19 | 40 |
| 12 | https://emidius.mi.ingv.it/ASMI/event/19950612_1813_000?page=2 | 1995 06 12 | Roman Countryside | 125 | 47 | 125 |
| 13 | https://emidius.mi.ingv.it/ASMI/event/19950824_1727_000?page=2 | 1995 08 24 | Pistoia Apennines | 56 | 53 | 217 |
| 14 | https://emidius.mi.ingv.it/ASMI/event/19951230_1522_000?page=2 | 1995 12 30 | Fermano | 106 | 6 | 114 |
| 15 | https://emidius.mi.ingv.it/ASMI/event/19961015_0955_000?page=2 | 1996 10 15 | Emilian Plain | 135 | 125 | 768 |
| 16 | https://emidius.mi.ingv.it/ASMI/event/19970512_1350_000?page=2 | 1997 05 12 | Martani Mountains | 57 | 29 | 381 |
| 17 | https://emidius.mi.ingv.it/ASMI/event/19970902_1042_000?page=2 | 1997 09 02 | Zafferana Etnea | 33 | 17 | 40 |
| 18 | https://emidius.mi.ingv.it/ASMI/event/19971111_1844_000?page=2 | 1997 11 11 | Etna_S.Maria | 35 | 16 | 41 |
| 19 | https://emidius.mi.ingv.it/ASMI/event/19971203_0828_000?page=2 | 1997 12 03 | Etna_Southwest Side | 6 | 7 | 24 |
| 20 | https://emidius.mi.ingv.it/ASMI/event/19971224_0940_000?page=2 | 1997 12 24 | Etna_Southern side | 11 | 34 | 97 |
| 21 | https://emidius.mi.ingv.it/ASMI/event/19980110_0845_000?page=2 | 1998 01 10 | Etna_Southwest Side | 44 | 14 | 69 |
| 22 | https://emidius.mi.ingv.it/ASMI/event/19990707_1716_000?page=2 | 1999 07 07 | Frignano | 32 | 13 | 123 |
| 23 | https://emidius.mi.ingv.it/ASMI/event/19990805_1457_000?page=2 | 1999 08 05 | Etna_Southwest Side | 35 | 34 | 53 |
| 24 | https://emidius.mi.ingv.it/ASMI/event/20000311_1035_000?page=2 | 2000 03 11 | Aniene Valley | 214 | 32 | 216 |





| 25 | https://emidius.mi.ingv.it/ASMI/event/20010109_0251_000?page=2 | 2001 01 09 | Zafferana Etnea | 104 | 67 | 105 |
|----|----|----|----|----|----|----|
| 26 | https://emidius.mi.ingv.it/ASMI/event/20010422_1356_000?page=2 | 2001 04 22 | Etna_Western side | 55 | 15 | 70 |
| 27 | https://emidius.mi.ingv.it/ASMI/event/20010503_2141_000?page=2 | 2001 05 03 | Etna_Ragalna | 13 | 9 | 34 |
| 28 | https://emidius.mi.ingv.it/ASMI/event/20010713_0315_000?page=2 | 2001 07 13 | Etna_Southern side | 25 | 17 | 57 |
| 29 | https://emidius.mi.ingv.it/ASMI/event/20010714_0553_000?page=2 | 2001 07 14 | Etna_C.da Calcerana | 16 | 7 | 19 |
| 30 | https://emidius.mi.ingv.it/ASMI/event/20011028_0903_000?page=2 | 2001 10 28 | Etna_S. M. Ammalati | 67 | 20 | 86 |
| 31 | https://emidius.mi.ingv.it/ASMI/event/20020922_1601_000?page=2 | 2002 09 22 | Piano Provenzana | 35 | 10 | 67 |
| 32 | https://emidius.mi.ingv.it/ASMI/event/20021027_0250_000?page=2 | 2002 10 27 | Piano Provenzana | 17 | 54 | 106 |
| 33 | https://emidius.mi.ingv.it/ASMI/event/20021029_1002_000?page=2 | 2002 10 29 10 02 | Bongiardo | 38 | 66 | 151 |
| 34 | https://emidius.mi.ingv.it/ASMI/event/20021029_1639_000?page=2 | 2002 10 29 16 39 | Scillichenti | 7 | 43 | 65 |
| 35 | https://emidius.mi.ingv.it/ASMI/event/20021031_1032_000?page=2 | 2002 10 31 | Molise | 51 | 168 | 798 |
| 36 | https://emidius.mi.ingv.it/ASMI/event/20030126_1957_000?page=2 | 2003 01 26 | Forlì Apennines | 35 | 21 | 184 |
| 37 | https://emidius.mi.ingv.it/ASMI/event/20030213_0532_000?page=2 | 2003 02 13 | Piano Pernicana | 4 | 18 | 73 |
| 38 | https://emidius.mi.ingv.it/ASMI/event/20030411_0926_000?page=2 | 2003 04 11 | Scrivia Valley | 78 | 108 | 741 |



| 39 | https://emidius.mi.ingv.it/ASMI/event/20030914_2142_000?page=2 | 2003 09 14 | Bologna Apennines | 134 | 84 | 692 |
|----|------|------|------|------|------|------|
| 40 | https://emidius.mi.ingv.it/ASMI/event/20040601_1032_000?page=2 | 2004 06 01 | Piano Pernicana | 17 | 18 | 85 |
| 41 | https://emidius.mi.ingv.it/ASMI/event/20041124_2259_000?page=2 | 2004 11 24 | Western Garda | 176 | 265 | 1575 |
| 42 | https://emidius.mi.ingv.it/ASMI/event/20050822_1202_000?page=2 | 2005 08 22 | Lazio Coast | 58 | 14 | 316 |
| 43 | https://emidius.mi.ingv.it/ASMI/event/20051031_0002_000?page=2 | 2005 10 31 | Trecastagni | 32 | 15 | 139 |
| 44 | https://emidius.mi.ingv.it/ASMI/event/20060520_0705_000?page=2 | 2006 05 20 | Etna_Southwest Side | 27 | 12 | 168 |
| 45 | https://emidius.mi.ingv.it/ASMI/event/20061219_1458_000?page=2 | 2006 12 19 | Etna_Northwest Side | 28 | 23 | 170 |

**Authors contribution**
AT designed the research and led the discussions. All the co-authors wrote the initial paper and edited
all the following versions. All the co-authors contributed to the datasets compilation. AA made all the
figures and the dataset elaborations.
**Competing interest**
The contact author has declared that none of the authors has any competing interests.
**Acknowledgements**

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
