# Peer review of "A comprehensive integrated macroseismic dataset from 1"

_Earth System Science Data, 2025_

## Author Comment (AC2)

We appreciate Referee 1's positive feedback on our paper. It is our intention to extend our methodology to other earthquakes for which multiple studies are available in CTPI15. Below are the responses to each of the questions raised by Referee 1.

- You might cite a couple of recent papers discussing the effects of the application of MCS and EMS98 scales to macroseismic magnitudes.

- R: Ok we inserted Vannucci et al 2021 citation in the Introduction.

- Online questionnaires are cited in the introduction but are not mentioned further. Please explain why you believe they are not useful to integrate the macroseismic datasets.

- R: This first attempt at integration of datasets did not include earthquakes assessed through online questionnaires data. This kind of integration involves more complex analysis, which we are addressing in separate work anyway.

- In case studies, please report and compare available instrumental magnitudes with macroseismic ones.

- R: ok done, we inserted a sentence for each case study. As far as the October 27, 2002 concerns we would like to point out that the macroseismic magnitude obtained by D'Amico et al. (2025) is calibrated with the local magnitude (Ml) therefore not directly comparable with Mw magnitude.

- It could be interesting to compare the average difference between macroseismic and instrumental magnitudes and between magnitudes computed from MCS and EMS98 intensities before and after the data integration operation.

R: We inserted in the paper further analyses at the end of Section 6, as following:

Having access to a large dataset of intensity data in both the EMS-98 and MCS scales has also allowed us to carry out a comparative analysis of macroseismic magnitude estimates derived from the two scales. To this end, we calculated the macroseismic magnitude for each selected event (Appendix A) using intensity data expressed in both the EMS-98 and MCS scales. For the 19 earthquakes that occurred in the volcanic region of Mt. Etna, macroseismic magnitudes were estimated using the most recent relationship developed by D'Amico et al. (2025), whereas the remaining 26 events were calculated using the algorithm proposed by Gasperini et al. (1999, 2010). As expected, the comparison between magnitudes based on EMS-98 and MCS data shows no significant differences: the average difference between the macroseismic magnitudes derived from the two scales is small, amounting to - 0.01 units. In detail, Figure 8 shows that the differences between macroseismic magnitudes estimated using EMS-98 and MCS data do not exceed 0.2 magnitude units.

[Figure]

**Figure 8: Comparison between macroseismic magnitudes based on EMS-98 and MCS data.**

Figure 9 shows the comparison between the macroseismic estimates derived from the dataset introduced in this study and the instrumental magnitudes available from the CPTI15 catalogue. This analysis has been performed only for tectonic earthquakes as for the volcanic events the macroseismic magnitude was obtained by the calibration with the local magnitude (Ml) (D'Amico et al. (2025), not directly comparable with Mw magnitude.

Although some differences are observed, the average difference between the instrumental moment magnitude and the macroseismic magnitude is minimal, with an average difference of - 0.05 units.
In contrast, Figure 10 shows the comparison between instrumental magnitudes and the macroseismic estimates currently included in the CPTI15 catalogue, which exhibit a larger average difference of -0.17 units. These results highlight a significant improvement achieved through the revised dataset proposed in this study. Figure 11 shows the differences of each event between instrumental magnitudes reported in CPTI15 and two sets of macroseismic magnitudes: those derived in this study (blue dots) and those reported in CPTI15 (red dots) . We excluded the earthquakes that occurred in the Etna region from this analysis.

[Figure]

Figure 1: Comparison between the macroseismic magnitude obtained with our dataset in the EMS-98 scale and the instrumental magnitude reported in CPTI15.

[Figure]

Figure 2: Comparison between the macroseismic magnitude and the instrumental magnitude reported in CPTI15.

[Figure]

Figure 3: Residuals between the instrumental magnitudes of CPTI15 and macroseismic magnitudes obtained in this study (blue dots) and those reported in CPTI15 (red dots). The x-axis reports the event IDs as listed in Appendix A.

Gasperini et al. (1999, 2010) are cited but not listed in References.

- R: Thank you, we inserted it in the References list

---

## Author Response (AR1)

**RC1**: 'Comment on essd-2025-10'

The paper is very interesting and well written. I have a long experience (as a user) of the Italian macroseismic database and in the past, I always hoped for the integration of macroseismic data from different studies instead of choosing only one of them as preferred. Finally this important operation has been carried out in a thoughtful and accurate way. I suggest extending further such integration to all remaining earthquakes for which multiple studies are available.

I have only few minor comments for the authors:

- You might cite a couple of recent papers discussing the effects of the application of MCS and EMS98 scales to macroseismic magnitudes.

- R: Ok we inserted Vannucci et al 2021 in the Introduction.

- Online questionnaires are cited in the introduction but are not mentioned further. Please explain why you believe they are not useful to integrate the macroseismic datasets.

- R: This first attempt at integration of datasets did not include earthquakes assessed through online questionnaires data. This kind of integration involves more complex analysis, which we are addressing in separate work anyway.

- In case studies, please report and compare available instrumental magnitudes with macroseismic ones.

- R: ok done, we inserted a sentence for each case study. As far as the October 27, 2002 concerns we would like to point out that the macroseismic magnitude obtained by D'Amico et al. (2025) is calibrated with the local magnitude (Ml) therefore not directly comparable with Mw magnitude.

- It could be interesting to compare the average difference between macroseismic and instrumental magnitudes and between magnitudes computed from MCS and EMS98 intensities before and after the data integration operation.

- R: We inserted a comparison between macroseismic magnitudes computed from MCS and EMS98 intensities at the end of Section 6.

  Figure 1 shows the comparison between the macroseismic estimates derived from the dataset introduced in this study and the instrumental magnitudes available from the CPTI15 catalogue. Although some differences are observed, the average difference between the instrumental moment magnitude and the macroseismic magnitude is minimal, with an average difference of - 0.05 units. In contrast, Figure 2 shows the comparison between instrumental magnitudes and the macroseismic estimates currently included in the CPTI15 catalogue, which exhibit a larger average difference of -0.17 units. These results highlight a significant improvement achieved through the revised dataset proposed in this study. Figure 3 shows the differences of each event between instrumental

magnitudes reported in CPTI15 and two sets of macroseismic magnitudes: those derived in this study (blue dots) and those reported in CPTI15 (red dots) . We excluded the earthquakes that occurred in the Etna region from this analysis.

We believe that this comparison falls outside the scope of this work, which primarily aims to describe the new datasets. Therefore, we prefer not to include this analysis in the manuscript.

[Figure]

Figure 1: Comparison between the macroseismic magnitude obtained with our dataset in the EMS-98 scale and the instrumental magnitude reported in CPTI15.

[Figure]

Figure 2: Comparison between the macroseismic magnitude and the instrumental magnitude reported in CPTI15.

[Figure]

Figure 3: Residuals between the instrumental magnitudes of CPTI15 and macroseismic magnitudes obtained in this study (blue dots) and those reported in CPTI15 (red dots). The x-axis reports the event IDs as listed in Appendix A.

Gasperini et al. (1999, 2010) are cited but not listed in References.

- R: Thank you, we inserted it in the References list

**RC2**: 'Comment on essd-2025-10', Ludmila Provost, 18 Apr 2025  reply
The paper presents updated macroseismic fields for 45 earthquakes that occurred in Italy by integrating different existing studies. The authors present the methodology used to integrate the different studies. The input data of the different studies are clearly described and the methodology used is well outlined. The case studies presented illustrate the methodology applied, the difficulties encountered by the authors and how they resolved them. The paper as a whole is well written, and thanks to this, the methodology applied can be easily reproduced for other earthquakes and also in other area. As an occasional user of the Italian database, I was pleased to see the improved macroseismic fields updated in this study, and hope that this work will be extended to other earthquakes.

I have one minor comment, which is more a question for the authors:

In your methodology guidelines, you stated that "Localities with intensity value (I) in the EMS-98 and MCS scales assigned after a field survey have been included in the new

dataset without further check". In the case study of the Moline earthquake, if I understand well, you did reassess intensity for a locality for a locality that had been assigned an intensity value after a field survey. I understand that it was necessary, but did you have a specific reason to check again the data associated with this locality?

Due to the fact that the direct survey was performed in the MCS scale only, we had to get more data to assess the EMS-98 intensities also. New information allowed us to rebuild a more comprehensive scenario that resulted in modified MCS intensities for 2 localities: San Giuliano di Puglia (from 8-9 MCS to 8 MCS, and 8 EMS ) and Petrella Tifernina (from 5-6 MCS to 6 MCS and 6 EMS). We acknowledge that in a few isolated cases, we have deviated from our guidelines, as new data, including coeval images, convinced us that the previous intensity assignments were not accurately reflecting reality.